# A2B Adenosine Receptor in Idiopathic Pulmonary Fibrosis: Pursuing Proper Pit Stop to Interfere with Disease Progression

**DOI:** 10.3390/ijms24054428

**Published:** 2023-02-23

**Authors:** Wiwin Is Effendi, Tatsuya Nagano

**Affiliations:** 1Department of Pulmonology and Respiratory Medicine, Faculty of Medicine, Universitas Airlangga (UNAIR), Surabaya 60132, Indonesia; 2Department of Pulmonology and Respiratory Medicine, Universitas Airlangga Teaching Hospital, Surabaya 60015, Indonesia; 3Pulmonology and Respiratory Medicine of UNAIR (PaRU) Research Center, Universitas Airlangga Teaching Hospital, Surabaya 60015, Indonesia; 4Division of Respiratory Medicine, Department of Internal Medicine, Graduate School of Medicine, Kobe University, 7-5-1 Kusunoki-cho, Chuo-ku, Kobe 650-0017, Japan

**Keywords:** purinergic signaling, adenosine, A2BAR, chronic respiratory diseases, idiopathic pulmonary fibrosis

## Abstract

Purine nucleotides and nucleosides are involved in various human physiological and pathological mechanisms. The pathological deregulation of purinergic signaling contributes to various chronic respiratory diseases. Among the adenosine receptors, A2B has the lowest affinity such that it was long considered to have little pathophysiological significance. Many studies suggest that A2BAR plays protective roles during the early stage of acute inflammation. However, increased adenosine levels during chronic epithelial injury and inflammation might activate A2BAR, resulting in cellular effects relevant to the progression of pulmonary fibrosis.

## 1. Introduction

In response to tissue injury, the cell releases various factors responsible for local inflammation and activates myofibroblasts for wound repair. In the context of routine wound healing, this process ends with myofibroblast apoptosis and inflammation reduction. However, pathological responses to tissue injury may turn into fibrosis and diseases.

Lung fibrosis is the consequence of excessive connective tissue that leads to structural transformations of the lung architecture. Fibrotic lung disease represents a vast spectrum of pulmonary pathologies characterized by different degrees of inflammation and fibrosis. The highest degree of fibrosis in these disorders is idiopathic pulmonary fibrosis (IPF). IPF is a chronic lung disease of unknown etiology characterized by the activation of fibroblasts and myofibroblast differentiation [1,2,3]. Growing evidence indicates that purinergic signaling is involved in the pathogenesis of IPF; however, its detailed role is not entirely known.

Purines and their derivatives, most notably adenosine diphosphate (ADP) and adenosine triphosphate (ATP), regulate intracellular energy homeostasis and nucleotide synthesis. Burnstock designed two main types of purinergic receptors, i.e., P1 and P2 [4]. Receptors for adenosine were classified as P1, while ATP and ADP were more suitable as natural ligands for P2 [5]. Based on the latest nomenclature of the International Union of Pharmacology Committee on Receptor Nomenclature and Drug Classification (NC-IUPHAR), the receptor for adenosine is named adenosine receptor (AR), which can be subdivided into four types: A1, A2A, A2B, and A3 [6].

Adenosine, released from injured or ischemic tissues, is vital in accelerating wound healing and tissue repair [7]. Both A2A and A2B contribute to regular wound repair and fibrosis in organ-specific ways, and interestingly, A2BAR may have opposite effects in different organs [8]. A2BAR-driven angiogenesis provides oxygen and nutrients required to support the growth and function of damaged tissues [9]. Recent studies reported that adenosine signaling pathways mediate inflammation and tissue remodeling via chemotaxis of neutrophils and fibroblasts in pulmonary fibrosis [10,11]. Elevations of A2BAR have modulated fibrosis in multiple organs [12,13,14].

IPF is an urgent health problem with an estimated incidence range of 3–9 cases per 100,000/year in Europe and North America, yet, is more minor in East Asia and South America [15]. There is currently no cure for IPF. The primary purpose of treatment is to reduce the symptoms and delay the progression. The precise role of A2BAR in IPF is still enigmatic. In this review, we examine the roles of purinergic signaling, with emphasis on A2BAR as an alternative treatment for IPF.

## 2. Role of A2BAR in IPF

Fibrosis is characterized by an accumulation in fibroblast proliferation, overproduction of extracellular matrix (ECM) proteins, and the formation of myofibroblasts that express α-smooth muscle actin (α-SMA). Fibrosis arises due to the dysregulation of the wound healing process at either the proliferative or remodeling stages or if the irritant persists continuously, driving the pathological process [16]. IPF, characterized by the pro-fibrotic epithelial–fibroblast interactions and progressive accumulation of ECM, results from aberrant wound-healing processes that lead to fibrosis [17].

The histopathological and/or radiological hallmark of IPF is usual interstitial pneumonia (UIP). UIP is characterized by heterogeneous areas of normal-appearing lung intermixed with older sub-pleural and paraseptal fibrosis, honeycombing pattern, and ECM-producing myofibroblasts termed fibroblast foci (FF) [18,19]. UIP of IPF is a mixture of acute and chronic histopathological fibrosis that could predict prognosis by FF activity and inflammation [20].

Fibrosis in IPF is initially suggested to result from a vigorous interstitial inflammatory response to an unknown cause. However, pathology results did not display significant inflammation, and immunosuppression is less effective as a treatment [21]. It is known that repetitive epithelial microinjury triggers the early development of pulmonary fibrosis. Chronic microinjury leads to an anomaly of programmed cell death, uncontrolled myofibroblast activation, excessive coagulation cascade, and progressive deposition of ECM [22]. The role of inflammation is still controversial, yet, pro-inflammatory cytokines and innate and adaptive immune responses accompany all fibrosis stages [23].

One of the most potent inflammation regulators is adenosine. It controls the function of inflammatory cells via interaction with specific receptors expressed on mast cells, leucocytes, neutrophils, eosinophils, and macrophages [24]. Following the concept of retaliatory metabolites, increases in the level of extracellular adenosine promote the healing process after inflammation-induced injury [25]. In acute states, adenosine plays primarily beneficial roles; however, the accumulation of adenosine levels beyond the acute injury phase can become dangerous by activating pathways that promote tissue injury and fibrosis [26]. Adenosine released at injured alveolar epithelial cells may modulate chronic inflammation-induced fibrosis.

Adenosine is a nucleoside molecule that regulates various physiological responses of the human body via ARs on the membrane surface of specific cells or tissues (Figure 1). Intracellular adenosine concentrations result from a mismatch between ATP synthesis and during ischemia, injury, or hypoxia. While extracellular adenosine originates from the active transport of intracellular or the breakdown of nucleotides outside the cell [27].

Each AR mediates the biological functions of extracellular adenosine. The four receptor subtypes have been purified and successfully cloned from mammalian and non-mammalian species, particularly rats, mice, and humans [28]. Adenosine receptors, members of heteromeric guanine nucleotide-binding protein (G protein)-coupled receptor (GPCR), have a seven-pass transmembrane a-helical structure with an extracellular amino terminus and an intracellular carboxyl terminus. Further, N-terminal domain with N-glycosylation sites influences the trafficking of the receptor to the plasma membrane [29].

A2BAR has been cloned from the rat hypothalamus, human hippocampus, and mouse mast cells and found in the organs such as the bowel, bladder, lung, and vas deferens [30]. A2BARs are low-affinity receptors (about 1000 nM) [31]; therefore, A2BAR delivers poor physiological relevance in comparison with the other ARs [32]. Their sensitivity can be increased by interaction with PKC to make them potential triggers of multiple signaling cascades [33]. Under injury, hypoxia, or cell stress, high expression of A2BAR will upgrade the level of adenylate cyclase (AC) activity, cAMP production, and protein kinase A (PKA) phosphorylation [31].

In the lungs, A2BAR is expressed on most inflammatory cells and has pro- and anti-inflammatory functions [34]. It generates anti-inflammatory effects by coupling with protein Gs and pro-inflammatory effects by coupling with protein Gq [35]. Its anti-inflammatory actions are intricately linked with wound healing, tissue regenerative, and fibrotic processes [36]. A recent study demonstrated that adenosine synthesis and expression of A2BAR in pulmonary fibrosis and emphysema were increased [37].

Although other cell types certainly make significant contributions, epithelial, fibroblasts, and alveolar macrophages are the most crucial drivers in the progression of pulmonary fibrosis. Because A2BAR interacts with the immune and inflammatory cells, we propose several potential mechanism actions of A2BAR in the pathogenesis of IPF.

### 2.1. A2BAR Induces Pro-Inflammation and Pro-Fibrotic Factors

Due to its characteristics as a low-affinity AR, A2B has an enigmatic function. Among other adenosine receptors, A2BAR was predominantly found in human lung epithelial cells [38,39]. The role of A2BAR as pro- or anti-inflammatory in epithelial lung injury may depend on several factors, including type and duration of damage, the concentration of adenosine produced, cytokines, and the interaction between A2BAR and other ARs. Zhou et al. described that genetic removal of the A2BAR during acute and chronic stages of lung injury determines the development of pulmonary fibrosis [40].

The direct elevation of A2BAR in acute injury/inflammation is to protect the lung epithelial barrier (tissue-protective). In a mice model of bleomycin-induced acute lung injury (ALI), A2BAR protected and maintained epithelial integrity via occludin and phosphorylated FAK (p-FAK) [41]. A2BAR agonists resolve ALI spontaneously by enhancing pulmonary cAMP levels and alveolar fluid clearance (AFC) [42]. Further, deletion of A2BAR was associated with a more severe degree of ALI due to enhanced loss of barrier function and increased pulmonary inflammation [43].

In the setting of an acute reaction, A2BAR can be beneficial as an anti-inflammatory by regulating multiple immune cells and various chemokines [44]. A slight elevation of A2BAR is essential to promote tissue-protective responses in acute epithelial injury. Recent studies support the protective role of A2BAR. Upregulation of A2BAR attenuated inflammatory response and reduced oxidative stress-associated apoptosis in hepatic ischemia-reperfusion injury [45]. Further, A2BAR activation produced a protective effect in cardiac ischemia-reperfusion injury via reduced apoptosis rate and ROS level [46]. However, a contrasting result was demonstrated as A2BAR mediated inflammation during lung ischemia-reperfusion injury in mice models [47].

It has been established that epithelial injury and loss of integrity of the alveolar epithelium are central to the pathogenesis of IPF. The development of IPF results from unsuccessful acute inflammation resolution to eliminate etiological factors and restore altered tissues.

A great deal of evidence supports the involvement of A2BAR in chronic inflammation responses. As opposed to its protective effect in the acute setting, overproduction of A2BAR in chronic inflammation is associated with organ fibrosis (tissue-destructive functions) [48]. Previously, Sun and colleagues for the first time verified the role of A2BAR in chronic lung diseases, acting as pro-inflammatory and pro-fibrotic factors [49]. There was an abundant expression of A2BAR in the lung tissue IPF and other chronic lung diseases [50].

A2BAR generates the production of numerous pro-inflammatory and pro-fibrotic mediators. Activation of A2BAR upgraded levels of hyaluronan synthases (HAS) and hyaluronan, a glycosaminoglycan that contributes to pulmonary fibrosis [37]. Moreover, a study showed that A2BAR signaling elevated pro-inflammatory mediators, including IL-6 and IL-8, in lung IPF patients [51]. Accumulation of adenosine activates A2BAR to produce transforming growth factor-β (TGF-β) and vascular endothelial growth factor (VEGF) [52]. Therefore, genetic deletion of A2BAR decreases inflammation and reduces lung fibrosis [53].

Persistent microinjury in IPF patients triggers chronic inflammation and high production of adenosine. Elevated adenosine activates all the receptor subtypes in order of decreasing affinity. However, some of them could be desensitized during long lasting activation, while activation of A2B is more involved in the production of fibrotic factors. Constant epithelial microinjury provokes a chronic inflammatory process leading to end-stage fibrotic scarring. Therefore, understanding the benefit or detrimental role of A2BAR in adenosine-based therapies for acute and chronic diseases is crucial. How A2BAR is involved in the pathogenesis of IPF is illustrated in Figure 2.

### 2.2. A2BAR Modulates Epithelial–Mesenchymal Transition and Apoptosis

Epithelial–mesenchymal transition (EMT) is a physiological process needed for homeostasis and wound healing [54]. However, a recent hypothesis stated that aberrant epithelial and epithelial–mesenchymal cross-talk underlies the pathogenesis of IPF [55,56,57]. Recurrent microinjuries induce epithelial apoptosis and drive epithelial cells to transdifferentiate into EMT, in which alveolar epithelial cells undergo the transition to fibroblasts [58,59]. Vice versa, fibroblasts provide a pro-fibrotic environment that modulates epithelial apoptosis [60]. Although the precise mechanism remains unknown, various factors may induce the reprogramming of epithelial cells to become vulnerable and sensitive to apoptosis [61]. Dysregulation of EMT is associated with multiple pathological processes.

Many cancer and organ fibrosis studies have reported the correlation between the accumulation of adenosine levels and the EMT process. Extracellular nucleotides and adenosine promote EMT in organ fibrosis in the heart, liver, lung, and renal epithelial cells [62]. In the same way as inflammation, adenosine signaling in cancer can play multifaceted functions, with anti-tumor or pro-tumor responses [63]. However, many studies in cancer models propose a dominant pro-tumor activity. A recent review by Alvarez et al. described that adenosine plays a vital role in EMT [64]. Further, adenosine signaling modulated EMT via the cAMP-dependent PKA pathway in breast cancer [65,66]. In contrast, extracellular adenosine inhibits the migration and invasion of cervical cancer cells by suppressing the EMT progress [67].

Previous studies described the correlation between A2AAR and EMT in organ fibrosis [68,69]. However, there is only little evidence of the A2BAR subtype regulating EMT. In the pathogenesis of IPF, A2BAR plays an essential role in modulating EMT. The activation of A2BAR may mitigate EMT by controlling the ERK pathway. A2BAR decreased the expression of epithelial markers (E-cadherin) and enhanced mesenchymal markers (N-cadherin) through the activation balance of TGF-β1-independent cAMP/PKA and MAPK/ERK pathways in human epithelial lung cells [70]. Recently, inhibition activation of A2BAR downregulated matrix metalloproteinase-9 (MMP-9) activity and EMT expression in glioblastoma stem-like cells [71].

Of the four adenosine receptors, A2AAR and A3AR are the most vital in the modulation of cell death; however, activation of A2BAR regulates apoptosis. The plasticity role of A2BAR during acute settings is to protect tissue by alleviating apoptosis. Purinergic signaling suppresses apoptotic cells via A2BAR in fin regeneration [72]. Acute activation of A2BAR may help counteract ischemia and acute injury. Indeed, direct injury-induced apoptosis is the main element of the earlier wound-healing response [73]. The protective role of A2BAR-associated apoptosis was through a cAMP-dependent pathway [74]. In the ALI mouse model, A2BAR suppresses lung epithelial apoptosis by modulating ERK, p38, and JNK phosphorylation [75]. Further, the interaction between hypoxia-inducible factor-1α (HIF-1α) and A2BAR alleviated inflammation and apoptosis in hepatic ischemia-reperfusion injury [45].

In contrast to acute injury, high levels of adenosine-associated apoptosis might have been implicated in the progression of chronic disease. Pulmonary fibrosis is characterized by increased apoptosis of alveolar epithelial cells and decreased apoptosis of fibroblasts. Therefore, increased apoptosis of epithelial cells leads to inefficient re-epithelialization and, conversely, apoptosis resistance of fibroblasts leads to increased fibrosis [76]. A2BAR activation inhibits apoptosis resulting in the proliferation and development of multiple cancer cells and solid tumors. High expression of A2BAR in colorectal carcinomas was related to low apoptosis and cancer cell growth [77]. A2BAR stimulated cell proliferation and declined apoptosis in prostate cancer [78].

Furthermore, Gallardo et al. found that A2BAR proliferated breast cancer cells via the adenylate cyclase/PKA/cAMP signaling pathway [79]. Therefore, blockade of A2BAR ameliorates tumor development of bladder, breast, and melanoma cells [80,81]. However, other studies showed that the mechanism of A2BAR is to inhibit apoptosis instead of inducing apoptosis. A2BAR induces cell cycle arrest and apoptosis to reduce tumor growth via the ERK1/2 cascade and mitochondrial signaling pathway [82,83].

The specific mechanisms of apoptosis-associated A2BAR in the development and progression of pulmonary fibrosis have yet to be extensively and systematically explored. The pathways and factors identified that lead to pathological apoptosis in organ fibrosis and cancer are almost identical. The cellular and molecular mechanisms in the pathogenesis of organ fibrosis and cancer are similar [84]. Considering that A2BAR can induce and inhibit apoptosis in cancer cells, A2BAR might also regulate apoptosis in IPF. A recent study emphasized the novel role of the A2BAR in regulating programmed cell death during the resolution of inflammation and tissue repair [85].

From all the work on A2BAR associated with EMT, we suggested mechanisms for how A2BAR triggers aberrant EMT responses in IPF patients. Adenosine concentrations may increase following epithelial injury. Initially, short-term activation of A2BAR controls inflammation to attenuate fibrosis by maintaining EMT balance. A2AAR also reduces inflammation-mediated lung fibrosis. However, the persistent microinjury induces the elevation of adenosine significantly, resulting in more A2BAR being activated. Chronic A2BAR signaling may promote lung fibrosis. A2BAR might have different roles during the acute and chronic stages of lung fibrosis.

### 2.3. A2BAR Regulates the Differentiation of Fibroblasts

Fibroblasts are tissue mesenchymal cells that regulate the production of ECM. In the normal wound healing process, after fibroblast activation followed by myofibroblast differentiation, fibroblasts undergo apoptosis to prevent ECM overproduction and scar formation [86]. In IPF, fibroblasts tend to be apoptosis-resistant and become immortal. A recent study demonstrated that fibroblast apoptosis resistance occurs earlier and precedes scar formation [87].

The role of A2BAR is contradictory during acute and chronic lung injury. A2BAR is an essential link between hypoxia and adenosine signaling in acute lung injury. Hypoxia-dependent signaling pathways and factor HIF-1α induce the expression of A2BAR in dampening hypoxia-induced inflammation [88]. A2BAR attenuated inflammation in endotoxin-induced acute lung injury [89]. They deliver a decisive anti-inflammatory role by enhancing Tregs [90]. Further, A2BAR protects against lung injury via IL-10 expression, not neutrophil infiltration [91]. A recent study demonstrated that A2BAR inhibits systemic inflammatory response and alleviates lung injury through the NF-κB pathway [92]. Activation of A2BAR leads to accumulation of adenylate cyclase-dependent cAMP and increased PKA activity [42]. Moreover, A2BAR-associated cAMP regulating epithelial sodium channel (ENaC) can alleviate pulmonary edema in the LPS-induced ALI model [93].

While A2BAR diminishes inflammation in acute lung injury, A2BAR may enhance the production of various inflammatory cytokines and pro-fibrotic mediators in chronic lung injury. It has been proposed that A2BAR mediates fibroblast proliferation and promotes myofibroblast differentiation [94]. A2BAR induced differentiation of fibroblasts into myofibroblasts via IL-6, especially during hypoxia [95]. In a recent study, A2BAR also promoted fibroblast proliferation and myofibroblast differentiation via follistatin-like 1 (Fstl1) and TGF-β [14]. Vasiukov and colleagues suggested that A2A or/and A2B adenosine receptors regulate the TGF-β axis in the functions of fibroblasts [96]. As a consequence, the blockade of A2BAR decreased myofibroblast marker expression [97].

Indeed, the activation of A2BAR is essential in regulating acute and chronic lung disease. Zhou et al. demonstrated that A2BAR supports anti-inflammatory factors to diminish acute lung injury but acts as a pro-fibrotic factor in developing pulmonary fibrosis [40]. Perhaps, the differential role of A2BAR arises from the diverse impacts of downstream signaling in acute or chronic lung injury [98].

### 2.4. A2BAR and Macrophage Polarization

Pulmonary macrophages are classified into monocyte-derived alveolar (AMs) and tissue-resident alveolar or interstitial macrophages (IMs). By their vast phenotypic plasticity, macrophages can be polarized to different cell phenotypes as either classically (M1) or alternatively activated (M2), defined by their capacity to induce inflammatory or anti-inflammatory immune responses, respectively [99]. However, whether AM exhibits an apparent dichotomy of polarization M1/M2 needs to be better defined. In contrast, IMs are less well-studied.

In IPF, macrophage heterogeneity regulates the development of pulmonary fibrosis from the early phases of injury and the fibrotic phase. Monocyte-derived AM expressed higher pro-inflammatory and pro-fibrotic functions and persisted after the resolution of lung injury and fibrosis; therefore, depletion of monocyte-derived AM ameliorates lung fibrosis [100]. In concordance, AM pyroptosis, polarization, apoptosis, and interaction with lung epithelial cells contribute to the development of lung fibrosis [101].

Adenosine is involved in macrophage differentiation, maturation, proliferation, and polarization in response to cellular stress and damage [102]. Littlefield et al. showed that M1 phenotype is associated with the upregulation of the A2AAR and A2BAR [103]. In addition, A2AAR attenuates inflammation via downregulating M1 activation [104]. In contrast, stimulatory effects of adenosine in suppressing inflammation via upregulating M2 activation are mediated predominantly by A2BAR [105]. Interestingly, a recent study revealed that A2BAR has no effect in regulating macrophage polarization and differentiation [106].

The effect of adenosine on human macrophages has yet to be fully understood. Indeed, A2BAR might function as an “anti-inflammatory” or “pro-inflammatory” depending on points in the progression of inflammation [44]. Following an acute vascular injury, activation of A2BAR suppressed TNF-α production by macrophages [107]. Deletion of A2BAR in a mice model demonstrated increased liver inflammation [108]. Further, A2BAR inhibitors suppressed the expression of inflammatory mediators and chemokines in the early stage of renal injury [109]. Moreover, A2BAR limits adipose tissue inflammation via upregulating IL-4-associated cytokines such as CCAAT enhancer-binding protein-β, interferon regulatory factor 4, and peroxisome proliferator-activated receptor-γ [110].

The plasticity role of A2BAR-associated macrophages shows how complex each type of inflammation is. Following intratracheal bleomycin, A2BAR increases macrophage-related inflammation; however, chronic bleomycin exposure elevated the production of pro-fibrotic factors IL-6 from alveolar macrophages [40]. Another study revealed that A2BAR steered renal myofibroblast differentiation and overproduction of pro-inflammatory and pro-fibrotic mediators [111]. A2BAR signaling directed macrophages to the anti-inflammatory/pro-fibrotic M2 phenotype [112]. In addition, genetic deletion A2BAR in mice models showed an insufficient polarization towards M2 macrophages [113].

Furthermore, the interaction of HIF1α–A2BAR contributes to the development and progression of pulmonary fibrosis via the expression of M2, cell differentiation, and production of pro-fibrotic mediators [114]. Quintana et al. found that A2BAR pointing polarizes macrophages to a fibrotic M2 phenotype in a mice model of fibrosis [53]. Therefore, administration of A2BAR inhibited infiltration and activation of M1 but increased anti-inflammatory/pro-fibrotic activated macrophages M2 in renal fibrosis [109]. de Leve et al. emphasized that the deletion of CD37 and A2BAR diminished the accumulation of M2 during the development of pulmonary fibrosis [115].

The newest study demonstrates a significant role for A2BAR and Netrin-1 in promoting macrophage-associated lung fibrosis [116]. A2BAR generates macrophage polarization toward the M2 phenotype in cancer [25]. Furthermore, El-Naccache et al. showed that A2BAR drives a type 2 immune response characterized by the upregulation of M2 macrophages in helminth infection [117]. These results suggest that A2BAR is involved in the upregulation of macrophage M2.

## 3. Targeting the A2BAR Signaling Pathway as Therapy for Pulmonary Fibrosis

The pro-fibrotic role of the A2BAR has been supported by many studies which exhibited attenuation of interstitial fibrosis. GS-6201 is a selective, potent, and orally available A2BAR receptor antagonist that has been used to inhibit fibrosis. Quintana and his colleagues found that GS-6201 significantly reduced alternatively activated macrophages (M2) and the production of IL-6 in the mouse model of dermal fibrosis [13]. Further, administration of GS-6201 in a mice model could prevent caspase-1-related apoptosis and significantly alleviate cardiac remodeling after acute myocardial infarction [118]. Most in vivo studies have identified an anti-fibrotic role of selective A2BAR antagonist GS-6201 in organ fibrosis.

In line with its effect on other organ fibrosis, preclinical studies showed that GS-6201 or CVT-6883 significantly reduced elevated markers of inflammation, fibrosis, and pulmonary injury in vivo models. GS-6201 inhibited the progression of fibrotic and vascular lesions leading to pulmonary hypertension associated with endothelin-1 (ET-1) and IL-6 in interstitial lung diseases [119]. Recently, genetic deletion of A2BAR or inhibition by antagonist GS-6201 reduced levels of hyaluronan synthase 2 (Has2), IL-6, and transglutaminase 2 (Tgm2) [120]. CVT-6883 alleviated pulmonary inflammation and expression of pro-fibrotic factors, TGF-β1 and osteopontin (OPN), leading to diminished pulmonary fibrosis in a model of bleomycin-induced pulmonary injury [49]. Liu et al. revealed that CVT-6883 diminishes TGF-β1-mediated fibroblast proliferation and myofibroblast differentiation [14].

In a phase 1 clinical trial of a randomized, double-blinded, placebo-controlled, single ascending dose study in 24 healthy volunteers, CVT-6883 was safe and well tolerated, with no serious adverse events reported [121]. However, there needs to be further information regarding phase II and III clinical trial results. Further analyses to extend these clinical trials in pulmonary fibrosis are warranted.

Nevertheless, the good news was brought when a study of novel non-xanthine A2BAR antagonists, BAY-545, reduced pro-inflammatory and pro-fibrotic mediators in lung animal models [122]. Moreover, selective adenosine A2BAR antagonist, PSB-603, decreased inflammatory response and oxidative stress [123]. However, the slow progression of specific ligand development has hampered the discovery of the functional significance of the A2BARs [124].

## 4. Conclusions

In recent years the increasing knowledge about the role of adenosine and related ARs in the dysfunction pathways of chronic lung diseases has regarded them as a valuable drug target. A2BARs are paradoxical inflammatory modulators that demonstrate both anti-inflammation and pro-inflammation. These opposing roles depend on timing and concentration.

Short-term activation of the A2BAR receptors decreases inflammation. Due to being widely distributed throughout the body, A2BAR activates protection for tissues and cells against injury, hypoxia, and ischemia. However, excessive A2BAR in chronic conditions could be harmful. Persistent epithelial injury induces elevation of adenosine and activates A2BAR in maintaining chronic inflammation, modulating aberrant EMT, regulating myofibroblast differentiation, and polarization of macrophages, leading to lung fibrosis.

Understanding the involvement of A2BAR signaling pathways will provide novel and essential information into the potential role of A2BAR agonists or antagonists. A2BAR may have a bi-phasic effect on inflammation and fibrosis, and caution must be taken when deciding to treat patients with chronic lung diseases. For example, the A2BAR antagonist might be beneficial in treating patients with chronic lung diseases, such as IPF, only when it is given after the disease status is established. Therefore, the timing of A2BAR antagonist treatment to inhibit the progression of A2BAR-associated fibrosis in IPF is very substantial.

A2BAR might be the most adenosine-insensitive receptor fully activated only under chronic pathological conditions, such as IPF. Focusing on the appropriate time to give A2BAR inhibitors might be the most appealing pharmacological alternative for treating IPF. Therefore, additional studies in animal models and clinical trials are needed to investigate the right time to give A2BAR antagonists in treating pulmonary fibrosis.

## Figures and Tables

**Figure 1 ijms-24-04428-f001:**
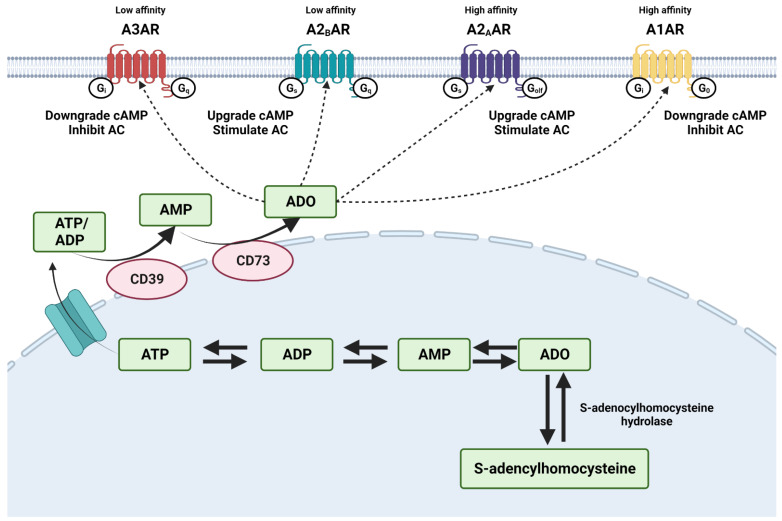
Production, transport, and metabolism of adenosine (ADO). Intracellular adenosine is produced via dephosphorylation from the primary source, AMP, and hydrolysis of S-adenosyl-homocysteine through the enzyme S-adenosyl-L-homocysteine hydrolase. Extracellular adenosine results from hydrolysis of ATP and ADP by CD39, dephosphorylation of adenosine monophosphate (AMP) by CD73, and from the active transport of intracellular. Extracellular adenosine binds four adenosine receptors (A1AR, A2AAR, A2BAR, and A3AR) on the surface of cells.

**Figure 2 ijms-24-04428-f002:**
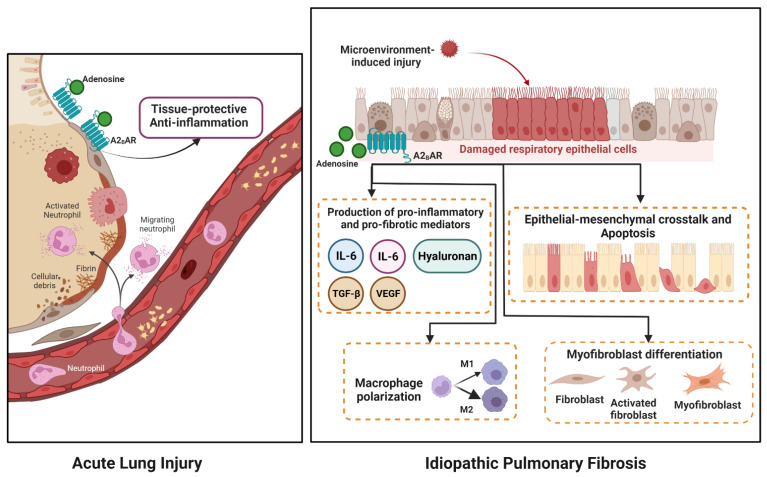
In the acute setting, adenosine through A2BAR acts as an anti-inflammatory function and delivers a tissue protective role with attenuation of pulmonary inflammation. In chronic lung diseases such as IPF, elevated extracellular adenosine activates A2BAR in preserving chronic inflammation. Persistent microinjury facilitates A2BAR in triggering aberrant EMT responses and epithelial apoptosis. A2BAR induces various pulmonary cell types to produce pro-inflammatory, pro-fibrotic mediators, including IL-4, IL-6, TGF-β, VEGF and hyaluronan synthetases, from macrophages and vascular smooth muscle cells. Furthermore, A2BAR signaling is responsible for activating fibroblast and myofibroblast differentiation. Signaling through A2BAR modulates macrophage that tends to M2 phenotype.

## Data Availability

Data available in a publicly accessible repository.

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
