# Peer review of "A2B Adenosine Receptor in Idiopathic Pulmonary Fibrosis: Pursuing Proper Pit Stop to Interfere with Disease Progression"

_ijms, 2023, doi:10.3390/ijms24054428_

Round 1
Reviewer 1 Report
The authors have undertaken a fairly comprehensive review of the roles of purinergic signaling with emphasis on A2BAR as an alternative treatment for idiopathic pulmonary fibrosis. Based on this, new potential lines of development are discussed. I did not have any major concerns, only several minor issues listed below:
Line 67, “express -smooth muscle actin (-SMA)”
Line 183, “factor- (TGF-)”
Line 200, “TGF-VEGF”
Line 293, “HIF-1”
Line 310, “TGF-”
Line 362, “HIF1A-A2BAR”
Author Response
Review report (reviewer 1)
The authors have undertaken a fairly comprehensive review of the roles of purinergic signaling with emphasis on A2BAR as an alternative treatment for idiopathic pulmonary fibrosis. Based on this, new potential lines of development are discussed. I did not have any major concerns, only several minor issues listed below:
Line 67, “express -smooth muscle actin (-SMA)”
Line 183, “factor- (TGF-)”
Line 200, “TGF-VEGF”
Line 293, “HIF-1”
Line 310, “TGF-”
Line 362, “HIF1A-A2BAR”
Author`s response
Thank you for your appreciation. We have already corrected several minor issues listed by reviewer 1 (using the track changes).

Reviewer 2 Report
The aim of this study was to summarize the main data through a short review on the possible role of the adenosinergic system (specially through A2R receptors) in the idiopathic pulmonary fibrosis.
The manuscript is of interest and well written.
I have however some remarks and suggestions.
Abstract : among the four adenosine receptors, the A2B are draws much attention because it has lowest affinity..
This sentence make no sense. I suggest :
Among the adenosine receptors, the A2B has the lowest affinity such that it was long considered to have little pathophysiological significance.
Introduction : no comment
2. Role of A2BAR in IPF : Line 67 what is @smooth ?
what is @-SMA ?
Line 98 : ..human body via AR should be ARs.
Line 114 A2BARs are low affinity receptors. Please give a range of affinity values.
See Beukers MW Mol Pharmacol 2000.
Line 116
Their affinity can be increased by interaction with PKC should be : their sensitvity can be increased…
Figure I : I suppose CD37 should be replaced by CD73.
2.1 Line 177 A2B AR generates nothing by itself.
I propose : Activation of A2B AR generates….
Line 187 : Elevated extracellular adenosine levels might predomnantly activate A2BAR more than other ARs to produce fibrotic factors.
This sentence is unclear. Elevated adenosine activates all the receptors subtypes in order of decreasing affinity. However some of them could be desentitized during long lasting activation while Activation of A2B is more involved in the production of fibrotic factors.
2.2 Line 235 MMP9 : this abbreviation is not precised.
2.4 Lines 341-342 Interestingly, a recent study revealed that A2BAR has no effect in regulating macrophage polarization…
because this fact is controversial and we can not conclude about the precise impact on macrophage polarization, in this context, I suggest to replace the title of the chapter 2.4 by : A2BAR and macrophage polarization
3 No comment
Conclusions is well done
Author Response
Review report (reviewer 2)
The aim of this study was to summarize the main data through a short review on the possible role of the adenosinergic system (specially through A2R receptors) in the idiopathic pulmonary fibrosis.
The manuscript is of interest and well written.
I have however some remarks and suggestions.
- Abstract : among the four adenosine receptors, the A2B are draws much attention because it has lowest affinity.
This sentence make no sense. I suggest : Among the adenosine receptors, the A2B has the lowest affinity such that it was long considered to have little pathophysiological significance.
- Introduction : no comment
- Role of A2BAR in IPF : Line 67 what is @smooth ? what is @-SMA ?
- Line 98: human body via AR should be ARs.
- Line 114: A2BARs are low affinity receptors. Please give a range of affinity values. See Beukers MW Mol Pharmacol 2000.
- Line 116. Their affinity can be increased by interaction with PKC should be : their sensitvity can be increased…
- Figure I : I suppose CD37 should be replaced by CD73.
- 1
- Line 177 A2B AR generates nothing by itself. I propose : Activation of A2B AR generates….
- Line 187 : Elevated extracellular adenosine levels might predomnantly activate A2BAR more than other ARs to produce fibrotic factors.
This sentence is unclear. Elevated adenosine activates all the receptors subtypes in order of decreasing affinity. However some of them could be desentitized during long lasting activation while Activation of A2B is more involved in the production of fibrotic factors.
- 2
- Line 235 MMP9 : this abbreviation is not precised.
- 4
- Lines 341-342 Interestingly, a recent study revealed that A2BAR has no effect in regulating macrophage polarization…
Because this fact is controversial and we can not conclude about the precise impact on macrophage polarization, in this context, I suggest to replace the title of the chapter 2.4 by : A2BAR and macrophage polarization
- 3 No comment
- Conclusions is well done
Author`s responses
Thank you for your suggestion. All the correction can be tracked (using the track changes).
- We modified based on suggestion reviewer`s
- We have already corrected several minor issues listed by reviewer, including figure 1
- We have already corrected several minor issues
- The abbreviation of MMP-9 is corrected
- Revision for sub chapter 2.4. (A2BAR and macrophage polarization)
